# An evaluation of prescribing trends and patterns of claims within the Preferred Drugs Initiative in Ireland (2011–2016): an interrupted time-series study

Ronald McDowell,[1] Kathleen Bennett,[2] Frank Moriarty,[1] Sarah Clarke,[3] Michael Barry,[4] Tom Fahey[1]

[1]HRB Centre for Primary Care Research, Department of General Practice, Royal College of Surgeons in Ireland Medical School, Dublin, Ireland
[2]Division of Population Health Sciences, Royal College of Surgeons in Ireland, Dublin, Ireland
[3]Health Service Executive Medicines Management Programme, Trinity Centre for Health Sciences, St James's Hospital, Dublin, Ireland
[4]National Centre for Pharmacoeconomics, Trinity Centre for Health Sciences, St. James's Hospital, Dublin, Ireland

**Correspondence to**
Dr Ronald McDowell;
ronaldmcdowell@rcsi.ie

## ABSTRACT

**Objective** To examine the impact of the Preferred Drugs Initiative (PDI), an Irish health policy aimed at enhancing evidence-based cost-effective prescribing, on prescribing trends and the cost of prescription medicines across seven medication classes.

**Design** Retrospective repeated cross-sectional study spanning the years 2011–2016.

**Setting** Health Service Executive Primary Care Reimbursement Service pharmacy claims data for General Medical Services (GMS) patients, approximately 40% of the Irish population.

**Participants** Adults aged ≥18 years between 2011 and 2016 are eligible for the GMS scheme.

**Primary and secondary outcomes** The percentage of PDI medications within each drug class per calendar quarter. Linear regression was used to model prescribing of the preferred drug within each medication group and to assess the impact of PDI guidelines and other relevant changes in prescribing practice. Savings in drug expenditure were estimated.

**Results** Between 2011 and 2016, around a quarter (23.59%) of all medications were for single-agent drugs licensed in the seven drug classes. There was a small increase in the percentage of PDI drugs, increasing from 4.64% of all medications in 2011 to 4.76% in 2016 (P<0.001). The percentage of preferred drugs within each drug class was significantly higher immediately following publication of the guidelines for all classes except urology, with the largest increases noted for lansoprazole (1.21%, 95% CI: 0.84% to 1.57%, P<0.001) and venlafaxine (0.71%, 95% CI: 0.15% to 1.27%, P=0.02). Trends in prescribing of the preferred drugs between PDI guidelines and the end of 2016 varied between drug classes. Total cost savings between 2013 and 2016 were estimated to be €2.7 million.

**Conclusion** There has been a small increase in prescribing of PDI drugs in response to prescribing guidelines, with inconsistent changes observed across therapeutic classes. These findings are relevant where health services are seeking to develop more active prescribing interventions aimed at changing prescribing practice.

## Strengths and limitations of this study

► Primary Care Reimbursement Service data cover pharmacy claims for prescriptions issued to General Medical Services (GMS) Scheme eligible patients (around 40% of the Irish population).
► Methods used are appropriate given the phased introduction of the preferred drug guidelines.
► GMS patients over-represent older adults and those in receipt of social welfare.
► The results based on aggregated data give an overview of the Preferred Drugs Initiative in its early years but require further detailed analysis to examine prescriber and patient heterogeneity.

pharmacists for the cost of prescription items issued to General Medical Services (GMS) eligible patients via the Primary Care Reimbursement Service (PCRS).[1] This is the largest community drug scheme in Ireland, providing access to free or minimal cost healthcare for patients whose household income falls below the eligibility threshold specified by the Irish Government, as well as the majority of people aged ≥70 years (approximately 95%) where a higher income threshold applies. Currently, GMS eligible patients in Ireland have their prescription charges paid directly by the State, with a patient-levy of €2.50 for each item dispensed, up to a maximum of €25 per month. Historically, Ireland has spent as much as 50% above the European Union average per capita on drugs for a variety of reasons, such as low levels of use of generic medications and higher negotiated prices with pharmaceutical companies for both patented and generic drugs.[2 3]

Against the background of an ageing population,[4] the economic downturn of 2008 and rising drug costs, the HSE established the Medicines Management Programme (MMP) in 2013. The MMP has undertaken a number

## BACKGROUND

The Health Service Executive (HSE) in Ireland spent €1.05 billion in 2015 reimbursing

of initiatives aimed at enhancing evidence-based and cost-effective prescribing,[5] one of which is the Preferred Drugs Initiative (PDI). The PDI recommends a single 'preferred drug' within a therapeutic drug class as the prescriber's drug of first choice. Factors considered when selecting the preferred drug include clinical efficacy, ease of administration, the possibility of side effects or interactions with other drugs, cost and national and international clinical guidelines. Recommendations for preferred drugs are made on an ongoing basis, with the findings disseminated through the publication of prescribing guidelines and General Practioner (GP) meetings. The regulations covering generic substitution of branded medications are separate to the PDI guidelines, with generic substitution of drugs implemented where possible unless there are clinical reasons for prescribing the branded medication. The issuing of preferred drugs is voluntary and no incentives are given to prescribers to issue the preferred drug instead of others from within the same therapeutic drug class, with the patient-levy remaining unaltered irrespective of preferred or non-preferred drug status. Although the preferred drug may not necessarily be the least expensive licensed medication within each drug class, it has been estimated that increased provision of the preferred drugs could save the HSE €15 million per year.[5]

As of September 2016, reports detailing the rationale behind the choice of the preferred drugs have been published for the first 10 therapeutic drug classes covered by the Initiative.[6] These are proton pump inhibitors (PPIs), statins, ACE inhibitors, angiotensin-II receptor blockers (ARBs), serotonin norepinephrine reuptake inhibitors (SNRIs), selective serotonin reuptake inhibitors (SSRIs), medications for treating urological conditions (urinary incontinence, frequency and overactive bladder), oral anticoagulants for stroke prevention in patients with non-valvular atrial fibrillation, beta-blockers and calcium channel blockers. There has been no evaluation of changes in prescribing following the introduction of the PDI until now. The aims of this paper are to: (i) examine the trends and patterns of pharmacy claims for seven PDI drug classes among eligible adult GMS patients in Ireland between 2011 and 2016; (ii) assess the impact of the PDI recommendations over time using segmented regression analysis and (iii) estimate the cost savings due to the PDI during these years.

## METHODS
The STrengthening the Reporting of Observational Studies in Epidemiology (STROBE) guidelines were used in the reporting of this study.[7]

### Data
HSE-PCRS monthly pharmacy claims were analysed from 2011 to 2016.[8] This study period provided an average of 3 years of claims data both before and after the PDI across the seven drug classes considered. The data include all pharmacy claims made for GMS patients and for which

the cost of the claim has been reimbursed to community pharmacies by the HSE.

### Preferred Drugs Initiative
The first seven medication classes covered by the PDI are considered in this paper. The preferred drugs in each of these classes were lansoprazole (PPIs), simvastatin (statins), ramipril (ACE inhibitors), candesartan (ARBs), venlafaxine (SNRIs), citalopram (SSRIs) and extended release (ER) tolterodine (urology medications). Guidelines for beta-blockers and calcium channel blockers were introduced in September 2016. Prescriptions issued to children (those under 18 years), hospital emergency items, out-of-hours prescriptions and items not considered medications (such as medical devices and dressings) were excluded; the PDI is primarily aimed at the treatment of adults in the general population.

### Analytical methods/approach
Descriptive statistics were used to summarise relevant medications from the HSE-PCRS database and the classes of PDI drugs. Only single-agent drugs are considered in this paper, as this is the primary focus of the PDI.

The timescale used for the analyses of time series depends on the research question of interest.[9] Calendar quarters (January–March, April–June, July–September and October–December) were used to aggregate the data consistent with other analyses of prescribing data using interrupted time series.[10–12] The use of calendar quarters was deemed clinically appropriate: changes in prescribing patterns tend to be gradual and guidelines are not necessarily disseminated or actioned on the first day of each calendar month. Furthermore, Irish GMS eligible patients in receipt of prescription medication can receive 3 months' worth of repeat prescriptions per consultation with their GP. For each therapeutic drug class, a linear regression model was used to estimate the percentage of the preferred drug per drug class per calendar quarter between 2011 and 2016, allowing for any changes that might have taken place following issuing of guidelines or other changes in clinical practice. This is a commonly used strategy for analysing interrupted time series.[13] For medicine groups where the only 'interruption' considered was dissemination of PDI guidelines, the regression equations used had the form

$$p_{ij} = (\beta_{0j} + \beta_{1j}x_{ij1} + \beta_{2j}x_{ij2} + \beta_{3j}x_{ij3}) + e_{ij} \qquad (i = 0, \ldots, 23)$$

where for each medicine group $j$ ($j = 1, \ldots, 7$)

$p_{ij}$ is the percentage of items of the preferred drug reimbursed at time (quarter) i.

$\beta_{0j}$ is the estimated percentage of items being preferred drugs at t=0 (Jan–Mar 2011).

$\beta_{1j}$ is the estimated change in the percentage of items being preferred drugs in the calendar quarter following guidelines (the 'change of level').

$\beta_{2j}$ is the estimated change in the percentage of items being preferred drugs per calendar quarter (the 'slope') before the guidelines.

$\beta_{3j}$ is the estimated change in the percentage of items being preferred drugs per calendar quarter (the 'slope') post-guidelines.

$e_{ij}$ is the residual for calendar quarter i.

The $x_{ijk}$ ($k = 1, 2, 3$) were calculated from the data according to standard practice.[14]

More than one change of level can be incorporated into any interrupted time series where this is relevant to the research question.[13 15] It was not feasible to include changes in the price of drugs in these models given the large number of drugs considered. Across the drug classes, all drugs were licensed and available in Ireland between 2011 and 2016 and all generics were licensed prior to the study period, the key exceptions being the licensing of generic duloxetine in March 2015 and the licensing of mirabegron in January 2013. These two events were incorporated into the analyses for SNRIs and urology medications, respectively.

Examination of the partial autocorrelation coefficients showed that there was significant residual autocorrelation between adjacent calendar quarters (but not between non-adjacent quarters) in each drug group, and this was incorporated into the models using Prais-Winsten regression.[16] The potential for seasonal autocorrelation was also considered: in this context, seasonal autocorrelation would mean that a given medication within a drug class is on average more or less likely to be prescribed than other drugs in the same class by virtue of the time of year. The PDI guidelines do not refer to any such clinical considerations[6] and we additionally hypothesised that seasonal autocorrelation would not be of statistical significance. This hypothesis was tested for each drug class by comparing the regression models which included Fourier terms to account for seasonality[9] and models without the seasonality terms. For each drug class, seasonal autocorrelation was not of statistical significance and the seasonality terms were removed on the grounds of parsimony.

The PDI guidelines were national guidelines and consequently no control groups were available with which to compare prescribing under the PDI. However, we constructed two reference groups using the drug classes beta-blockers and calcium channel blockers. These were drug classes for which PDI guidelines were launched in September 2016 (the preferred drugs being bisoprolol and amlodipine, respectively) but for which no recommendations had been made when the PDI guidelines were launched for the other drug classes. Given that the earlier guidelines were launched within 6 months of each other, two additional models were fitted: one examining prescribing of bisoprolol as the preferred beta-blocker over the study period, allowing for potential changes in prescribing when guidelines for PPIs/statins (April 2013) and SNRIs/SSRIs (April 2014) were disseminated, and one model examining prescribing of amlodipine as the preferred calcium channel blocker, allowing for potential changes in prescribing when guidelines for ACE inhibitors/ARBs (September 2013) and urology medications (October 2014) were issued.

By coincidence rather than design, issuing of guidelines for each medicine group occurred early in each of the calendar quarters listed above, with the exception of the guidelines for ACE inhibitors and ARBs. Sensitivity analyses were used to explore whether the results varied when the calendar quarters were constructed differently (March–May, June–August, September–November and December–February) for these groups. Given that the PDI guidelines were launched in phases, sensitivity analyses were also used to examine whether results were dependent on the length of time considered before and after guidelines.

The models above were used to estimate increases or decreases in costs for each drug group associated with the PDI. Where only one interruption to the time series was included in the model, the predicted number of preferred drug items from each class was compared with the number which would have been issued had the trend in prescribing estimated before the guidelines continued, that is, the estimates of $\beta_{0j}$ $(\widehat{\beta_{0J}})$ and $\beta_{2j}$ $(\widehat{\beta_{2J}})$ remained unchanged, the estimate of $\beta_{1j}(\widehat{\beta_{1j}})$ was constrained to be zero and the estimate of $\beta_{3j}(\widehat{\beta_{3J}})$ was set equal to $\widehat{\beta_{2J}}$. The difference in the number of preferred drug items under the two scenarios was multiplied by the average price of the preferred drug, calculated across all reimbursements between dissemination of the guidelines and the end of 2016. The difference in the number of non-preferred drug items was multiplied by a weighted average of the price of all other drugs from within the medicine class, weighted according to the overall distribution of these items between issuing of the guidelines and the end of December 2016. These two costs were combined to give an overall cost differential. The process was extended analogously to include multiple interruptions as appropriate.

All analyses were conducted using Stata14.0SE.[17] The results were held to be significant if they referred to statistical significance on a two-sided design-based test evaluated at the 5% level.

## RESULTS

### Descriptive statistics

A total of 336 535 263 prescription items for medications were reimbursed by 4465 PCRS prescribers for 1 919 681 GMS adults aged 18 years and over between 2011 and 2016. The median number of items reimbursed per GMS patient was 63 (IQR 13–246) with a median total cost per patient of €905.75 (IQR €170.25–€4109.38). Approximately 55 million items were reimbursed per year, with the number of items peaking slightly in 2012 and 2013. During the 6-year period, 48.8 million (19.86%) prescription items were for the single-agent medicines licensed across the seven therapeutic drug classes considered. The drug classes most commonly prescribed to GMS patients

**Table 1** Summary of impact of Preferred Drugs Initiative (PDI) (2011–2016)

| Preferred drug class | PPI | Statin | ACE inhibitor | ARB | SNRI | SSRI | Urology | Total |
|---|---|---|---|---|---|---|---|---|
| Total no. of items | 18 929 282 | 19 944 634 | 8 837 006 | 5 171 204 | 3 345 307 | 8 348 567 | 2 239 263 | 333 535 263 |
| % of all drugs | 5.63 % | 5.93 % | 2.63 % | 1.54 % | 0.99 % | 2.48 % | 0.67 % | 19.86 % |

| Preferred drug | Lansoprazole | Simvastatin | Ramipril | Candesartan | Venlafaxine | Citalopram | ER tolterodine |
|---|---|---|---|---|---|---|---|
| Total no. of single-agent items | 4 571 751 | 1 313 389 | 4 719 996 | 557 622 | 1 155 600 | 1 650 520 | 577 540 |
| % within class | 24.14 % | 6.59 % | 53.41 % | 10.78 % | 70.99 % | 19.77 % | 25.79 % |
| Rank within class pre-PDI | 2/5 | 4/5 | 1/10 | 5/8 | 1/2 | 2/6 | 1/9 |
| Rank within class post-PDI | 2/5 | 4/5 | 1/10 | 5/8 | 1/2 | 3/6 | 3/9 |
| Absolute change in proportion of preferred drug items: first 3 months post-PDI versus previous 3 months | ↑ +0.98% (P<0.001) | ↑ +0.30% (P<0.001) | ↑ 0.53% (P<0.001) | ↓ −0.01% (P=0.99) | ↑ 0.30% (P=0.08) | ↓ −0.09% (P=0.37) | ↓ −0.98% (P<0.001) |

ARB, angiotensin-II receptor blocker; ER, extended release; PDI, Preferred Drugs Initiative; PPI, proton pump inhibitor; SNRI, serotonin norepinephrine reuptake inhibitor; SSRI, selective serotonin reuptake inhibitor.

were statins (5.93% of all items) and PPIs (5.63%), with the least common being SNRIs (0.99%) and drugs for treating urological conditions (0.67%). The descriptive statistics for each PDI medication class over the 6-year period are outlined in table 1.

The percentage of items relating to the seven drug classes increased slightly from 19.57% in 2011 to 20.04% in 2016, with small changes observed in the volume of prescriptions issued per each PDI medicine group over this time. More detailed breakdowns of PDI medicine groups per calendar year and quarter are given in online supplementary appendix tables A1 and A2 and figure A1.

### Preferred Drugs Initiative

Within the seven PDI drug classes considered, 23.59% of all prescription items were for the named preferred drugs. However, there was considerable variation between PDI drug classes both in terms of ranking and percentage coverage of the preferred drug (see table 1). The most commonly prescribed preferred drug within the relevant drug class was venlafaxine, which comprised 70.99% of all SNRI prescriptions. This was followed by ramipril (53.41% of all single-agent ACE inhibitors), ER tolterodine (25.79% of urology items), lansoprazole (24.14% of PPIs), citalopram (19.77% of SSRIs), candesartan (10.78% of all single-agent ARBs) and simvastatin (6.59% of all single-agent statins). The ranking of the preferred drugs within classes varied from first (ACE inhibitors and SNRIs) to second-last (statins). There was a small but statistically significant increase over time in the percentage of all medications, which was for the PDI drugs, increasing from 4.64% in 2011 to 4.76% in 2016 (P<0.001).

### Impact of clinical guidelines

Comparing prescribing patterns within each medication class in the 3 months prepublication and postpublication of the PDI guidelines, there was a small increase in the proportion of preferred drugs in four drug classes (PPIs (P<0.001), statins (P<0.001), ACE inhibitors (P<0.001) and SNRIs (P=0.08)), little change in two other drug classes (ARBs (P=0.99) and SSRIs (P=0.37)) and a reduction in percentage terms in prescribing of the PDI agent ER tolterodine (P<0.001) (table 1). Two preferred drugs, citalopram and ER tolterodine, were ranked lower within their respective classes between issuing of the guidelines and the end of 2016 than before. Figure 1 illustrates the secular trends for preferred drugs across the PDI categories by calendar quarters between 2011 and 2016: plots of the actual percentage of preferred drug items within each drug group between 2011 and 2016 are given in online supplementary appendix figure A2.

Segmented linear regression showed changes over time in the prescribing of all preferred drugs (table 2). In all medicine groups except urology, there was evidence of significant increases in prescribing of the preferred drugs immediately following dissemination of the PDI guidelines. For three medicine groups, there was significant evidence of an increase in the percentage of preferred drug items in the calendar quarter following issuing of the guidelines (lansoprazole (1.21%, 95% CI: 0.84% to 1.57%, P<0.001); venlafaxine (0.71%, 95% CI: 0.15% to 1.27%, P<0.001); simvastatin (0.30%, 95% CI: 0.1% to 0.5%, P=0.01)) and small increases in prescribing of the preferred drug in subsequent quarters. The percentage of SNRI medications which were venlafaxine did not change significantly immediately following the licensing

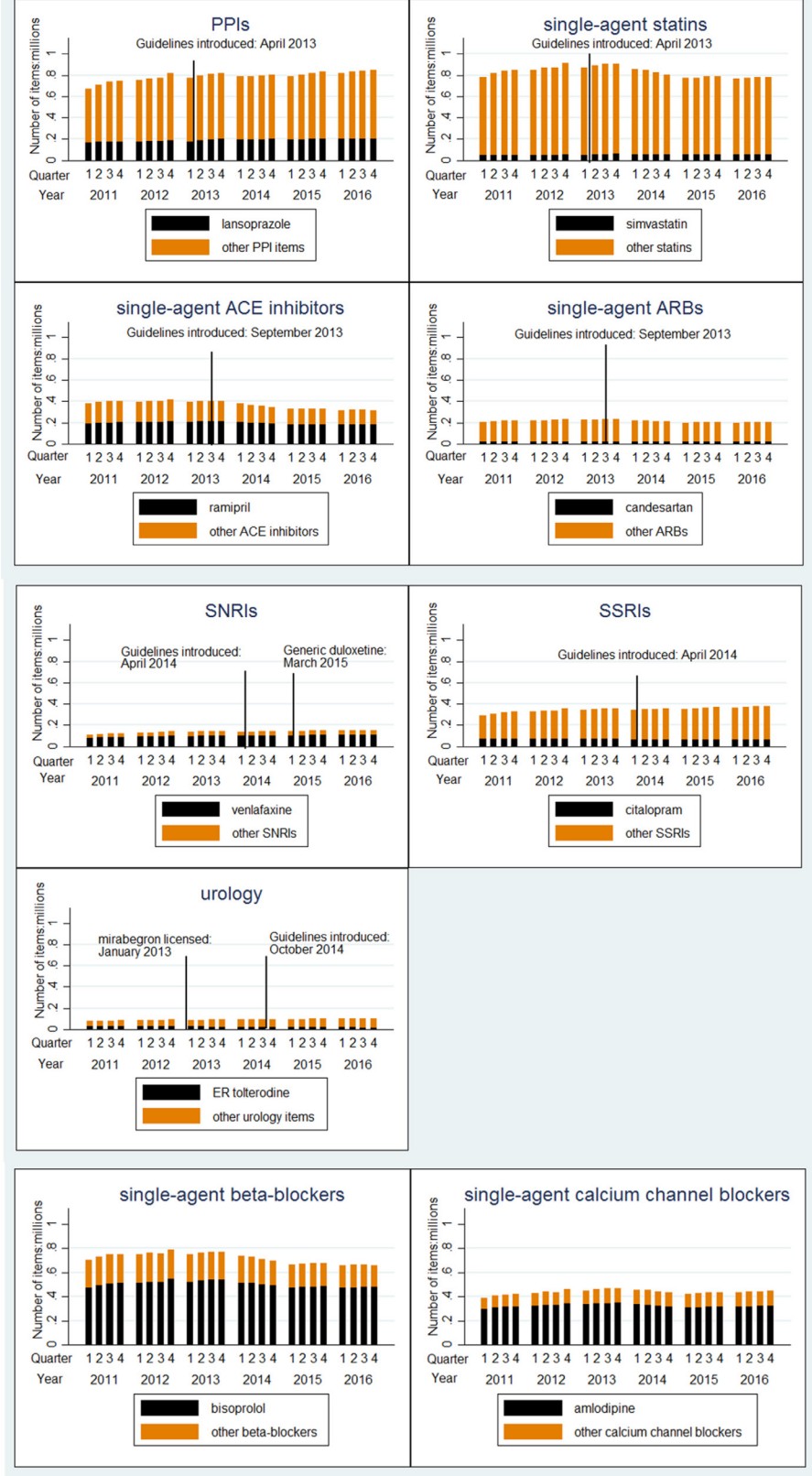

**Figure 1** Distribution of preferred drug items by therapeutic drug class. ARBs, angiotensin-II receptor blockers; ER, extended release; PPIs, proton pump inhibitors; SNRIs, serotonin norepinephrine reuptake inhibitors; SSRIs, selective serotonin reuptake inhibitors.

of generic duloxetine in March 2015 (P=0.76) or in subsequent quarters (P=0.34). For both candesartan and citalopram, for which prescribing within their PDI drug classes was in decline prior to the guidelines being issued, prescribing increased immediately following the PDI guidelines (candesartan (0.15%, 95% CI: 0.02% to 0.29%,

**Table 2** Segmented regression analysis in relation to PDI guideline publication, class-specific changes and cost savings

| Medicine group (preferred drug) | Guidelines introduced | Percentage of preferred drug items: Jan–March 2011 (SE), 95% CI | Increase in % of preferred drug items per quarter post March 2011 (SE), 95% CI, P value | Increase in % of preferred drug items Jan–Mar 2013 following licensing of mirabegron, (SE), 95% CI, P value | Increase in % of preferred drug items per quarter post March 2013 (SE), 95% CI, P value | Increase in % of preferred drug items in calendar quarter following PDI guidelines, (SE), 95% CI, P value | Increase in % of preferred drug items per quarter post PDI guidelines, (SE), 95% CI, P value | Increase in % of preferred drug April–June 2015 following introduction of generic duloxetine, (SE), 95% CI, P value | Increase in % of preferred drug items per quarter post June 2015, (SE), 95% CI, P value | Estimated savings between issuing of guidelines and Dec 2016 (€) |
|---|---|---|---|---|---|---|---|---|---|---|
| PPIs (lansoprazole) | April 2013 | 24.53 (0.47), (23.59 to 25.47) | −0.21 (0.05), (−0.32 to −0.11), P=0.001 | – | – | 1.21 (0.18), (0.84 to 1.57), P<0.001 | 0.04 (0.04), (−0.03 to 0.12), P=0.25 | – | – | 618 158 |
| Statins (simvastatin) | April 2013 | 5.94 (0.21), (5.50 to 6.38) | 0.02 (0.03), (−0.04 to 0.07), P=0.54 | – | – | 0.30 (0.10), (0.10 to 0.50), P=0.01 | 0.07 (0.02), (0.03 to 0.10), P=0.002 | – | – | 363 194 |
| ACE inhibitors (ramipril) | September 2013 | 49.14 (0.07), (48.99 to 49.28) | 0.38 (0.01), (0.35 to 0.40), P<0.001 | – | – | 0.16 (0.07), (0.01 to 0.31), P=0.04 | 0.41 (0.01), (0.39 to 0.42), P<0.001 | – | – | 50 163 |
| ARBs (candesartan) | September 2013 | 11.90 (0.08), (11.73 to 12.07) | −0.15 (0.01), (−0.17 to −0.12), P<0.001 | – | – | 0.15 (0.06), (0.02 to 0.29), P=0.03 | 0.01 (0.01), (−0.01 to 0.03), P=0.46 | – | – | 132 625 |
| SNRIs (venlafaxine) | April 2014 | 73.61 (0.44), (72.69 to 74.53) | −0.35 (0.05), (−0.46 to −0.24), P<0.001 | – | – | 0.71 (0.27), (0.15 to 1.27), P=0.02 | 0.26 (0.13), (−0.02 to 0.55), P=0.07 | −0.09 (0.30), (−0.73 to 0.54), P=0.76 | −0.08 (0.09), (−0.10 to 0.26), P=0.34 | 1 291 160 |
| SSRIs (citalopram) | April 2014 | 23.58 (0.13), (23.31 to 23.85) | −0.36 (0.01), (−0.39 to −0.33), P<0.001 | – | – | 0.30 (0.08), (0.12 to 0.47), P=0.002 | −0.23 (0.02), (−0.27 to −0.19), P<0.001 | – | – | 169 493 |
| Urology (ER tolterodine) | October 2014 | 37.27 (0.27), (36.69 to 37.84) | −1.00 (0.05), (−1.11 to −0.88), P<0.001 | 0.16 (0.24), (−0.35 to 0.66), P=0.52 | −1.04 (0.06), (−1.17 to −0.91), P<0.001 | −0.06 (0.24), (−0.57 to 0.45), P=0.82 | −0.63 (0.09), (−0.73 to −0.52), P<0.001 | – | – | 46 695 |
| Total savings | | | | | | | | | | 2 671 447 |

ARBs, angiotensin-II receptor blockers; ER, extended release; PDI, Preferred Drugs Initiative; PPIs, proton pump inhibitors; SNRIs, serotonin norepinephrine reuptake inhibitors; SSRIs, selective serotonin reuptake inhibitors.

P=0.03); citalopram (OR 0.30%, 95% CI: 0.12% to 0.47%, P=0.002)) but did not continue to increase significantly in subsequent quarters. Indeed, declines in prescribing of citalopram resumed in July 2014, although the decline was less steep than before the guidelines (P<0.001). There was a small increase in prescribing of the preferred ACE inhibitor (ramipril) immediately following the PDI guidelines (0.16%, 95% CI: 0.01% to 0.31%, P=0.04), although subsequent increases per calendar quarter did not differ significantly at the 5% level from increases observed per calendar quarter prior to the PDI guidelines (P=0.08). No statistically significant changes were observed in the prescribing of ER tolterodine immediately following the licensing of mirabegron in January 2013 (P=0.52) or the PDI guidelines in October 2014 (P=0.82), although the rate of decline in prescribing of ER tolterodine was lower following the PDI guidelines than between the licensing of mirabegron and dissemination of the PDI guidelines (P<0.001).

Sensitivity analyses showed that the results were materially unaffected when the calendar quarters used for analyses of ACE inhibitors and ARBs varied or when the length of time studied before and after the guidelines was changed (online supplementary appendix tables A3 and A4).

### Reference groups

Beta-blockers and calcium channel blockers accounted for 3.58% (n=12 056 378) and 2.30% (n=7 753 755) of single-agent medications for GMS patients between 2011 and 2016, with the most commonly prescribed medications being bisoprolol (56.83% of all single-agent beta-blockers (n=6 852 022)) and amlodipine (64.70% of all single-agent calcium channel blockers (n=5 016 348)), both of which were selected as preferred drugs in September 2016. There was a steady increase in prescribing of bisoprolol as the beta-blocker of choice and a consistent fall in prescribing of amlodipine within the calcium channel blocker medications over the study period. Effects in these drug groups associated with dissemination of the PDI guidelines for the other drug groups were non-significant at the 5% level (table 3). See figure 2 for plots of the estimated percentage of preferred drug items within each therapeutic drug class between 2011 and 2016.

### Cost savings

Overall, the cost savings after introduction of the PDI amounted to €2671k across all seven PDI drug classes (table 2). The savings associated with changes in prescribing following issuing of guidelines for the seven drug classes were estimated to be €123k in 2013, €396k in 2014, €837k in 2015 and €1314k in 2016. There were savings in each group, even though changes in dispensed medications were often minimal. The greatest impact was on the amount spent on SNRIs, with an estimated saving of €1291k between 2014 and 2016. This is due to the much higher cost of the non-preferred

**Table 3** Segmented regression analysis in relation to PDI guideline publication, reference groups

| Medicine group (preferred drug) | Guidelines introduced | Percentage of preferred drug items: Jan–March 2011 (SE), 95% CI | Increase in % of preferred drug items per quarter post March 2011 (SE), 95% CI, P value | Increase in % of preferred drug items per quarter post April–June 2013*, (SE), 95% CI, P value | Increase in % of preferred drug items per quarter post June 2013, (SE), 95% CI, P value | Increase in % of preferred drug items Oct–Dec 2013†, (SE), 95% CI, P value | Increase in % of preferred drug items per quarter post Dec 2013, (SE), 95% CI, P value | Increase in % of preferred drug items April–June 2014‡, (SE), 95% CI, P value | Increase in % of preferred drug items per quarter post June 2014, (SE), 95% CI, P value | Increase in % of preferred drug items Oct–Dec 2014§ (SE), 95% CI, P value | Increase in % of preferred drug items per quarter post Dec 2014, (SE), 95% CI, P value |
|---|---|---|---|---|---|---|---|---|---|---|---|
| Beta-blockers (bisoprolol) | September 2016 | 51.20 (0.03), (51.15 to 51.26) | 0.53 (0.01), (0.52 to 0.54), P<0.001 | −0.02 (0.05), (−0.13 to 0.09), P=0.71 | 0.50 (0.02), (0.45 to 0.54), P<0.001 | – | – | −0.05 (0.06), (−0.18 to 0.08), P=0.44 | 0.41 (0.001), (0.40 to 0.42), P<0.001 | – | – |
| Calcium channel blockers (amlodipine) | September 2016 | 68.18 (0.03), (68.12 to 68.29) | −0.34 (0.01), (−0.35 to −0.33), P<0.001 | – | – | 0.12 (0.06), (−0.001 to 0.23), P=0.06 | −0.26 (0.02) (−0.31 to −0.21), P<0.001 | – | – | 0.02 (0.07), (−0.13 to 0.17), P=0.76 | −0.21 (0.01) (−0.22 to −0.19), P<0.001 |

*Introduction of PDI guidelines for PPIs/statins.
†Introduction of PDI guidelines for ACE/ARBs.
‡Introduction of PDI guidelines for SNRIs/SSRIs.
§Introduction of PDI guidelines for urology medications.
ARBs, angiotensin-II receptor blockers; PDI, Preferred Drugs Initiative; PPIs, proton pump inhibitors; SNRIs, serotonin norepinephrine reuptake inhibitors; SSRIs, selective serotonin reuptake inhibitors.

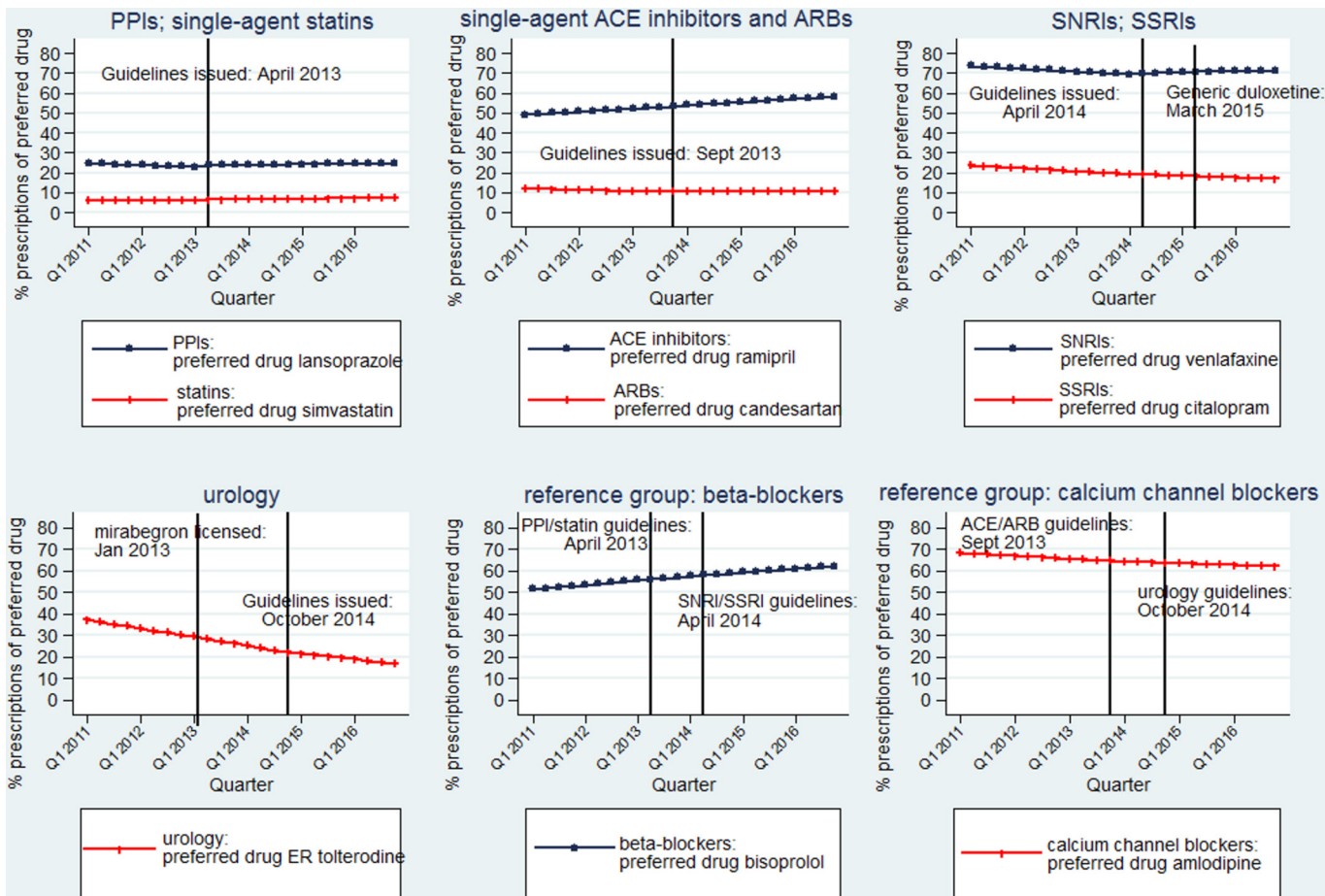

**Figure 2** Estimated percentage of preferred drugs by drug class: segmented regression models. ARBs, angiotensin-II receptor blockers; ER, extended release; PPIs, proton pump inhibitors; SNRIs, serotonin norepinephrine reuptake inhibitors; SSRIs, selective serotonin reuptake inhibitors.

drug duloxetine to the preferred drug venlafaxine. Other groups where the savings were marked were for the two larger volume groups where the guidelines had first been issued—PPIs saving €618k and statins saving €363k. For medicine groups where prescribing of the preferred drug was in decline before guidelines were issued, even the small short-term changes in prescribing translated into some savings. The smallest cost savings were in the prescribing of ramipril and ER tolterodine, due to the lack of change in prescribing trends observed within these groups between 2011 and 2016. The combined savings in the reference groups, had the prescribing patterns observed prior to the PDI guidelines remained unchanged, was an estimated €17k.

## DISCUSSION
### Principal findings
The seven drug classes considered that form part of the PDI accounted for approximately 20% of all medications reimbursed by the PCRS between 2011 and 2016. Changes in prescribing observed over the study period varied by PDI drug class, with substantial differences in the ranking order and quantity of preferred drug prescribed. Overall, the impact of the PDI guidance was limited, with

an inconsistent pattern observed across all therapeutic drug classes, and only a small increase (0.13%) in the percentage of preferred drugs issued overall between 2011 and 2016. Across the PDI drug classes. some differences emerged: in the first group of PDI drugs. there were increases in prescribing of the preferred drug immediately following issuing of the guidelines and continued though small increases subsequently (PPIs, statins and SNRIs); in the second group of PDI drugs (SSRIs and ARBs), there was a temporary increase in prescribing of the preferred drug just after the guidelines were issued and lastly, in the third group of PDI drugs (ACE and urology), there appeared to be little or no impact of clinical guidance. The reasons for such diversity are not known. ACE inhibitors are relatively inexpensive and this may account, in part, for the trend in ramipril prescribing remaining relatively unchanged. Although mirabegron has become the most commonly prescribed urology item since its launch in 2013, prescribing of ER tolterodine was in decline prior to this time.

### Context of other studies
PDI guidelines until now have been disseminated to prescribers mainly through correspondence and GP meetings. The literature shows that educational programmes

and publication of guidelines in themselves tend to have little effect on influencing prescribing practice, and that these need to be enhanced with other strategies.[18] In a systematic review of 79 studies examining interventions which changed doctor prescribing behaviour, the most effective interventions were patient-mediated interventions, outreach, audit and feedback and reminders.[19] In a study of changes in the use of losartan versus other single-agent ARBs in Sweden, investigators concluded that multiple and intensive demand-side measures are needed to change physician prescribing habits.[20] Other strategies which have been found to be helpful include direct involvement of the community pharmacist and face-to-face engagement from those seeking to encourage change with the prescriber.[21] Technological advances, such as alerts and prompts when issuing a drug may also prove useful.[22]

Any excess expenditure incurred through the issuing of non-preferred drugs to GMS patients is met directly by the HSE and not by the patient. Options which could reduce such expenditure include reducing choice for either patient or prescriber. It has been suggested that because prescribers can develop expertise of only a certain number of drugs, more restrictive formularies may also provide benefits to quality of prescribing.[23 24] In Sweden, the introduction of the 'Wise List', an evidence-based formulary of essential medicines, increased adherence to guideline recommendations in primary care from 80% to 90% and reduced variation in prescribing.[25] The introduction of co-payments, where the patient has to pay the difference between the price of the preferred drug and their chosen alternative, has the potential to be a considerable driver of change. Australia operates a therapeutic brand premium scheme, whereby a co-payment is required from patients when a prescriber has issued a drug within a drug class that is priced above the benchmark for drugs in that group.[26] While dramatic changes in co-payments may result in more patients switching to preferred agents (such as statins, ACE inhibitors and PPIs), they may also increase the risk of patients stopping their medication or becoming non-adherent.[27 28] Recent work has shown the drivers of drug expenditure in high-income countries vary substantially, with several other factors aside from physician prescribing behaviour and patient preference determining national drug expenditure.[29]

### Strengths and limitations

There are a number of strengths to this study. Our prescription sample is large and generalisable: PCRS data cover the entire GMS population of Ireland (around 40% of individuals). Despite the guidelines being introduced incrementally, the results were invariant to the time periods studied prepublication and postpublication of clinical guidelines. However, there are limitations to the study. GMS patients are weighted towards older adults and those socially and financially disadvantaged and so the results may not be reflective of the entire population in receipt of prescription medication. There is no way of knowing whether prescribers approached patients with regard to changes in their medication and/or whether these approaches were successful. Patient-specific factors may mean that issuing of the preferred drug may not have been appropriate or possible. Neither prescribers nor patients are homogeneous entities and considerable variation may exist within both. Although the changes in prescribing observed within the PDI medicine groups were not observed in the reference groups, there may be factors other than the PDI guidelines which have contributed to prescribing changes and the associated cost savings within the PDI drug groups.

### Policy implications and future research

The PDI has been developed to encourage evidence-based, cost-effective prescribing, but in view of the limited changes until now, it has delivered only a small amount of cost savings in terms of the money spent on these prescription items. If cost savings are to be maximised, the energies need to focus on medicine groups which are large volume (eg, PPIs and statins) and/or where there is considerable variation between the least and most expensive licensed medications in that group (eg, SNRIs). To enhance the impact of the PDI, multifaceted interventions appear most likely to succeed. Financial incentives to prescribers may be one possible component of such interventions, as operated in Irish primary care for a time in the 1990s[30]; however, any incentives for PDI drugs need to be aligned with professional values of prescriber and be mindful of personal preferences of patients taking long-term medication.[31–33] The effectiveness of such interventions is important to consider and although this has generally been evaluated using observational methods, experimental approaches may also be feasible.

Given the increasing demand for and costs associated with healthcare provision worldwide, the findings from this evaluation may be of interest to other countries seeking to provide treatment that is both evidence-based and cost-effective. This includes countries already implementing preferred drug schemes (eg, Australia), those which are considering such schemes or indeed any intervention aimed at changing clinical practice. The results show that initiatives which are primarily voluntary in nature may be impactful, but their impact can be limited and short term. They also show that interventions launched concurrently and developed using the same methodological framework may not necessarily yield similar results.

### CONCLUSIONS

Since the introduction of the PDI in 2013, there have been some cost savings across the PDI drug classes. However, more intensive implementation is needed if the PDI is to deliver the estimated €15 million per year cost saving that was anticipated. Multifaceted interventions will be required to enhance the coverage and impact of the PDI so that these benefits can be realised.

**Acknowledgements** Thanks to the Health Services Executive Primary Care Reimbursement Service (HSE-PCRS) for the use of pharmacy claims database.

**Contributors** RMcD drafted and planned all aspects of study design, cleaned and prepared data for analysis, conducted the statistical analyses and conducted a preliminary overview of the literature. KB prepared the monthly Primary Care Reimbursement Service claims downloads and gave significant methodological guidance on the analysis strategy. FM provided guidance on pharmaceutical matters and contributed to the discussion on context, policy implications and future research. SC and MB facilitated access to the claims data with the PCRS, gave detailed information on roll-out and implementation of the Preferred Drugs Initiative and contributed to interpretation of the results within the wider context of prescribing in Ireland. TF generated the research question and commented on the conduct, analysis and write-up of the paper. All authors read and approved submission of the paper.

**Funding** The primary author was funded by the Health Research Board (HRB) of Ireland under the HRB Centre for Primary Care Phase 2 Funding award, grant HRC/2014/1. KB was funded by HRB under grant RL-15-1579.

**Competing interests** None declared.

**Patient consent** Not required.

**Provenance and peer review** Not commissioned; externally peer reviewed.

**Data sharing statement** No additional data are available.

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
