## [Reviewer comments · BMJ Open]

ARTICLE DETAILS

TITLE (PROVISIONAL)	An evaluation of prescribing trends and patterns of claims within the Preferred Drugs Initiative in Ireland (2011-2016): an interrupted time-series study
AUTHORS	McDowell, Ronald; Bennett, Kathleen; Moriarty, Frank; Clarke, Sarah; Barry, Michael; Fahey, Tom

VERSION 1 – REVIEW

REVIEWER	Samantha Hollingworth School of Pharmacy University of Queensland Brisbane, Australia
REVIEW RETURNED	15-Sep-2017

GENERAL COMMENTS	This is an interesting and important study examining the (rather disappointing) effects of the PDI to improve the quality use of medicines and reduce medicine costs. The manuscript was well written and logical. A line spacing of at least 1.5 would have improved readability. Abstract Line 7 Consider one sentence in objective how the PDI reduces costs. (Intro: The PDI recommends a single 'preferred drug' within a therapeutic drug class as the prescriber's drug of first choice.) Abstract says GMS is 40% of the population but later (S&L) you say 38% - be consistent Introduction. Line 6 Give an estimate of the GMS population □ you can estimate a cost per person. Clarify if the preferred drugs are cheaper than others in the class. Otherwise how does the HSE estimate the savings? Does the PDI it incorporate branded vs generic prescribing? P32 no need to capitalise medicine groups (even though they make an acronym) Methods Line 11 Citation for data source Line 22 generic medicine names are not capitalised (ramipril) I am not able to comment on the statistical analyses.
---

	Results P7 line 18 (and other places) use the term 'proportion' (percentage is the measure) P8 line 3 no need for brackets around the 95%CI's P8 line 5 absolute increases - in prescriptions? Figures – consider modifying figures so that they are legible in black and white (i.e. white, grey, stippled, black bars, etc.). The blue, red, and green bars can be hard to differentiate, even in colour. Figure 2 – at least use different symbol type to differentiate. The line figures (e.g. 2A) are also hard to differentiate. Discussion P 9 line 49 Australia also has a therapeutic group premium https://www.pbs.gov.au/browse/group-premium Please clarify if patents would have paid more, less, or the same if changed to the preferred medicine. P10 line 42 What was the most widely used statin - atorvastatin? What was the price difference of simvastatin compared to other statins in the group? Would there have been more HSE savings if the preferred medicine had been prescribed more widely (6.6% is very low for the preferred one especially seeing as statins are the most widely prescribed class of medicines!).
--	---

REVIEWER	Amanj Kurdi Strathclyde Institute of Pharmacy and Biomedical Science University of Strathclyde, Scotland, UK
REVIEW RETURNED	24-Sep-2017

GENERAL COMMENTS	Comments Manuscript title: An evaluation of prescribing trends and patterns of claims within the Preferred Drugs Initiative in Ireland: 2011-2016 The study examined the impact of the Preferred Drugs Initiative (PDI), an Irish health policy aimed at reducing the cost of prescription medicines using segmented regression. Although the study looks interesting, it has major methodological limitations which make it unacceptable for publication at such. Background 1- Line 8- Not clear who are eligible for the PCR 2- Line 9- what does "means testing" mean? 3- Line 31- the authors mentioned that an evaluation report has been published in September 2016 for the first ten drug classes, but they did not mention the content of the report. It looks like the report has evaluated this PDI ; therefore, it is not clear what this study will add, i.e, I cannot see clear and strong justification and rationale for the study. Methods The study method suffers major limitations: Data Line 11- it is not clear how selecting 2011 to 2016 as the study
---

period will avoid confounding because other pharmaceutical policy changes could still happen within 2011-2016

PDI

Line 27- why the study population was limited to >18 years old? This is not suitable as the PDI targeted prescriptions which would be prescribed to any age group

Analytical method

1- The authors did not tested and accounted for auto-correlations as the study outcomes measured repeatedly each quarter are very likely to be auto correlated which will ultimately bias the study results
2- The analysis did not account for the impact of any other policies which might affect the study outcomes such as changes in drug prices, new treatment guidelines or when a drug in the class became off patent. For instance, the availability of generic candesartan, or change in hyperlipidemia management guidelines after the availability of generic atorvastatin which could have a potential impact on which drug to choose within the class.

3- Line 40- to perform segmented analysis properly, at least 12 time points are needed before and after the intervention to allow adjustment for seasonal autocorrelation; therefore, using calendar quarters to reduce the number of time-points is not appropriate.

4- The authors did not use any reference group as a control to ensure that the observed changes were potentially contributed to the PDI. I understand that the PDI was a national initiative and hence was not possible to identify an intervention free-area, however, other approaches could have been used such as using a related but different outcome within the intervention group that would not be influenced by the intervention.

Results

Through the whole result section, the authors described the changes as “modest” or “large” which is inappropriate. For instance, line 47, “a large reduction” to describe 0.98% reduction. All the changes in the first 3 months before and after the PDS described in Table 1 dose not even reach 1%.

Discussion

1- Again, inappropriate use of “modest” to describe the PDS impact

2- Line 21-24- here the authors show that factors other than the PDI were the reason behind the ACEIs and ER trends, and this could have been addressed, as I have explained earlier, by accounting for these other potential changes and by having a reference control group. Also, this can be further supported by the observed cost saving in ACEIs despite no change in prescribing trend after the PDI which explain that factors other than the PDI was the driving factor
3- It is well known from the literature that passive and voluntary initiatives are mostly ineffective, and that multiple interventions are usually required to influence prescribing behavior (line 29-60), and this was the authors' argument. But in fact the authors default contradicts their study findings by showing that PDI, which is a passive and voluntary initiative, has a significant impact of changing prescribing patterns and induce cost saving.

4- The authors mentioned the potential policy implications of their findings for Ireland, but cannot see any wider implications for the wider European and international community which would suggest the irrelevance of publishing this study in journals other than local Irish journals.

VERSION 1 – AUTHOR RESPONSE

Responses are in a separate word document: these are copied here below.

Responses to reviewers

Thank you to the editor and both reviewers for your helpful comments in the revision of this manuscript, and the opportunity to address these comments.

Editor Comments to Author:

- Please include the study design in the title. This is the preferred format of the journal.

Response: The title has been revised to: An evaluation of prescribing trends and patterns of claims within the Preferred Drugs Initiative in Ireland (2011-2016): an interrupted time-series study.

Reviewer: 1

R1: A line spacing of at least 1.5 would have improved readability.

Response: this has been changed to improve readability.

R1: Abstract Line 7 Consider one sentence in objective how the PDI reduces costs. (Intro: The PDI recommends a single 'preferred drug' within a therapeutic drug class as the prescriber's drug of first choice.)

Response: objective has been revised to read (p2): To examine the impact of the Preferred Drugs Initiative (PDI), an Irish health policy aimed at enhancing evidence-based cost-effective prescribing, on prescribing trends and the cost of prescription medicines across seven medication classes.

R1: Abstract says GMS is 40% of the population but later (S&L) you say 38% - be consistent.

Response: All figures have been revised to 40%.

R1: Introduction. Line 6 Give an estimate of the GMS population à you can estimate a cost per person.

Response: This has been now included on p10: The median number of items reimbursed per GMS patient was 63 (Interquartile Range (IQR) 13 to 246)) with a median total cost per patient of €905.75 (IQR €170.25 to €9,726.93).

R1: Clarify if the preferred drugs are cheaper than others in the class. Otherwise how does the HSE estimate the savings?

Response: Thank you for this comment: cost is one component to the recommendation of a preferred drug. Consequently the preferred drug may or may not necessarily be the cheapest drug within any drug class. We do not have access to the calculations used by the HSE whereby they obtained their projected savings. The text has been revised to say (p4):

Although the preferred drugs may not necessarily be the least expensive medications within each drug class, it has been estimated that increased provision of the preferred drugs could save the HSE €15 million per year.(5)

R1: Does the PDI incorporate branded vs generic prescribing?

Response: The issue of branded v generic prescribing is separate to the PDI. To-date reference prices have been set for some but not all generic drugs, including medicine groups not covered by the PDI. If a GMS patient wishes to receive a branded drug where a generic drug is available, they will be required to pay the price between the reference price and actual cost unless the prescriber has written "Do not substitute" on the prescription. The text has been amended to read (p4):

The regulations covering generic substitution of branded medications are separate to the PDI guidelines, with generic substitution of drugs implemented where possible unless there are clinical reasons for prescribing the branded medication.

R1: P32 no need to capitalise medicine groups (even though they make an acronym)

Response: we have corrected this in the text and graphs as appropriate

R1: Methods

Line 11 Citation for data source

Response: A citation for this has now been included.

R1: Line 22 generic medicine names are not capitalised (ramipril)

Response: This has been amended accordingly.

I am not able to comment on the statistical analyses.

R1: Results

P7 line 18 (and other places) use the term 'proportion' (percentage is the measure).

Response: Where percentages have been reported in the text or plotted on a graph, the text has been amended to percentage. The revisions of the models so that the outcome is percentage means that percentage is the appropriate scale on which to report results.

R1: P8 line 3 no need for brackets around the 95% CIs

Response: The formatting of the confidence intervals has been amended as follows:
lansoprazole (1.21%, 95%CI: 0.84% to 1.57%, p<0.001);

R1: P8 line 5 absolute increases - in prescriptions?

Response: As the outcome is now percentage coverage of PDI drug per class per quarter, the text refers to changes in the percentage coverage of the preferred drug within the class ie an increase in coverage from 20% to 21% is an increase of 1 %.

R1: Figures – consider modifying figures so that they are legible in black and white (i.e. white, grey, stippled, black bars, etc.). The blue, red, and green bars can be hard to differentiate, even in colour. Figure 2 – at least use different symbol type to differentiate. The line figures (e.g. 2A) are also hard to differentiate.

Response: We have revised the figures accordingly-Stata doesn't allow for striped/patterned bars in its settings.

R1: Discussion P 9 line 49 Australia also has a therapeutic group premium
<https://www.pbs.gov.au/browse/group-premium>

Response: We have inserted a sentence to this effect (p14);
Australia operates a therapeutic brand premium scheme, whereby a co-payment is required from patients when a prescriber has issued a drug within a drug class that is priced above the benchmark for drugs in that group.(23).

R1: Please clarify if patents would have paid more, less, or the same if changed to the preferred medicine.

Response: GMS patients in Ireland have their prescription charges paid directly by the State, with a patient-levy of €2.50 for each item dispensed, up to a maximum of €25 per month. This is the same regardless of whether it is a preferred or non-preferred drug. We have inserted in the text (p4)

Currently GMS patients in Ireland have their prescription charges paid directly by the State, with a patient-levy of €2.50 for each item dispensed, up to a maximum of €25 per month.

The issuing of preferred drugs is voluntary and no incentives are given to prescribers to issue the preferred drug instead of others from within the same therapeutic drug class, with the patient levy remaining unaltered irrespective of preferred- or non-preferred drug status.

R1: P10 line 42 What was the most widely used statin - atorvastatin? What was the price difference of simvastatin compared to other statins in the group? Would there have been more HSE savings if the preferred medicine had been prescribed more widely (6.6% is very low for the preferred one especially seeing as statins are the most widely prescribed class of medicines!).

Response: The most commonly prescribed statin currently in Ireland is atorvastatin (57.3% of all single-agent Statins between 2011-2016). simvastatin is the cheapest Statin, rosuvastatin the most expensive. The amount of savings will depend both on any increased volume of the preferred drug and the price differential between the preferred drug and what it has been substituted for. We have clarified this in the text as follows (p15):

If efforts are to be enhanced, the energies need to focus on medicine groups which are large volume (e.g. PPIs and statins) and/or where there is considerable variation between the least and most expensive licensed medications (e.g. SNRIs).

Reviewer: 2

Reviewer Name: Amanj Kurdi

Institution and Country: Strathclyde Institute of Pharmacy and Biomedical Science, University of Strathclyde, Scotland, UK Please state any competing interests or state 'None declared': None to declare

Please leave your comments for the authors below See the attached file please
Comments

Manuscript title: An evaluation of prescribing trends and patterns of claims within the Preferred Drugs Initiative in Ireland: 2011-2016

The study examined the impact of the Preferred Drugs Initiative (PDI), an Irish health policy aimed at reducing the cost of prescription medicines using segmented regression. Although the study looks interesting, it has major methodological limitations which make it unacceptable for publication at such.

R2: Background

1- Line 8- Not clear who are eligible for the PCR

2- Line 9- what does "means testing" mean?

Response: The text has been revised for clarity as follows (p4):

This (ie the PCRS GMS scheme) is the largest community drug scheme in Ireland, providing access to free or minimal cost health care for patients whose household income falls below the eligibility threshold specified by the Irish Government, as well as the majority of people aged ≥ 70 years (approximately 95%) where a higher income threshold applies.

R2: 3- Line 31- the authors mentioned that an evaluation report has been published in September 2016 for the first ten drug classes, but they did not mention the content of the report. It looks like the report has evaluated this PDI; therefore, it is not clear what this study will add, i.e, I cannot see clear and strong justification and rationale for the study.

Response: individual reports detailing the rationale for the choice of preferred drug within each class have been issued on an ongoing basis between April 2013 and Sept 2016, but to date no single report has been published which seeks to evaluate changes in prescribing which may have occurred with implementation of the PDI. Consequently this is the rationale for this research. The text has been revised to clarify this as follows (p5)

As of September 2016 reports detailing the rationale behind the choice of preferred drugs have been published for the first ten therapeutic drug classes covered by the Initiative.

An additional sentence has been added.

There has been no evaluation of changes in prescribing following the introduction of the PDI to date.

R2 Methods

The study method suffers major limitations:

Data

Line 11- it is not clear how selecting 2011 to 2016 as the study period will avoid confounding because other pharmaceutical policy changes could still happen within 2011-2016

Response. Thank you for this comment, this was a poorly worded and incorrect sentence. 2011-2016 was chosen as the study period so that across the 7 medicine groups as a whole the average time studied pre-PDI guidelines would be similar to the average studied post-PDI guidelines. We have revised this sentence as follows (p6):

This study period provided an average of three years of claims data both before and after the guidelines across the medication groups considered.

PDI

Line 27- why the study population was limited to >18 years old? This is not suitable as the PDI targeted prescriptions which would be prescribed to any age group.

Responses: The majority of the medication classes considered (e.g. statins, ACE, ARBs) are not commonly prescribed to children. The PDI was developed specifically to enhance cost-effective evidence-based prescribing among adults. The text has been clarified to say (p6):

; the PDI is primarily aimed at the treatment of adults in the general population.

Analytical method

R2- The authors did not test and account for auto-correlations as the study outcomes measured repeatedly each quarter are very likely to be auto correlated which will ultimately bias the study results

Response. Thank you for this comment which was a very important consideration. Autocorrelation coefficients were examined which showed there was significant autocorrelation at one lag only, and so the models were re-estimated to include this. In doing so we switched from logistic to linear regression, given that the denominators (number of drugs per class per quarter) are large. The text states this as follows (p7):

Examination of the autocorrelation and partial autocorrelation coefficients showed that there was significant residual autocorrelation between adjacent calendar quarters (but not between non-adjacent quarters) in each drug group, and this was incorporated into the models using Prais–Winsten regression (16).

R2- The analysis did not account for the impact of any other policies which might affect the study outcomes such as changes in drug prices, new treatment guidelines or when a drug in the class became off patent. For instance, the availability of generic candesartan, or change in hyperlipidemia management guidelines after the availability of generic atorvastatin which could have a potential impact on which drug to choose within the class.

Response:

We have considered this and extended two medicine groups as detailed below in the text (p7):

More than one change of level can be incorporated into any interrupted time series where this is relevant to the research question (13, 15). It was not feasible to include changes in the price of drugs in these models given the large number of drugs considered. Across the drug classes all drugs were licensed and available in Ireland between 2011 and 2016, and all generics were licensed prior to the study period, the key exceptions being the licensing of generic duloxetine in April 2015 and the

licensing of mirabegron in January 2013. These two events were incorporated into the analyses for SNRIs and urology medications respectively.

R2- Line 40- to perform segmented analysis properly, at least 12 time points are needed before and after the intervention to allow adjustment for seasonal autocorrelation; therefore, using calendar quarters to reduce the number of time-points is not appropriate.

Response:

We have considered this matter very carefully. All time series have the potential to exhibit seasonal correlation, however this does not mean that all time-series have to be analysed using month as the unit of time just to address seasonal autocorrelation and other time scales are possible. Wagner et al.(2002) states that that time series are required to summarise outcomes “at regular, evenly spaced intervals”. Consequently when they state (p305) “Detecting seasonality requires baseline series that span enough periods to detect these cyclic patterns...To estimate seasonal autocorrelation, the auto-regression model needs to evaluate correlations between error terms separated by 12 months” we interpret their sentence “Accounting for seasonally correlated errors usually requires at least 24 monthly data points” not to mean that data has to be analysed on a monthly time scale, but rather, where data is analysed on a monthly time scale (as is the case in the data they are presenting and analysing), there will be a full calendar year of data before the interruption and a full calendar year of data afterwards. We appreciate this sentence could be read either way.

Other papers make the matter more obvious: Bernal et al. (2017) state that methods for controlling for seasonality can include: “ a model stratified by the calendar month (or any other period)”. Bhaskan et al. (2013) show how to construct Fourier series to include seasonality in a time-series, “by creating a degrees variable for time divided by the number of time points in a year”. Therefore we understand the literature to say that the important feature is there is a full-calendar year pre/post any interruption, rather than that data has to be analysed on a monthly basis. Outcomes may not necessarily be expected to vary by month and they could vary by season.

The literature is not consistent regarding the number of time points to be included in interrupted time-series pre/post changes of level. Wagner et al.(2002) recommends 12 points on either side of the interruption, but go on to say that this is not based on estimates of statistical power but relates to having 12 monthly measures (i.e. a full calendar year) pre/post the interruption due to concerns over seasonality. Bernal et al. (2010) states that “power increases with the number of time points, but it is not always preferable to have more data points where historical trends have changed substantively”. They do not state a minimum number of points pre/post interruptions and suggest inspection of the data to get a visual impression on any trends. Flodgren & Oddgard-Jensen (2013) state you need “ at least three data points before and three data points after the intervention”.

In Penfold et al.'s teaching paper on interrupted time series (2013) there are ten quarterly time-points before the intervention and nine afterwards. We have at least nine time points on either side of the PDI guidelines in each drug group where there was only one change of level. Where a second change of level was incorporated as per the reviewer's helpful recommendations, we have still met Flodgren & Oddgard-Jensen's recommendation above. All our time-series were subject to sensitivity analysis to explore whether changing the number of time-points before and after the interruptions resulted in any substantive changes to our results: they did not.

Our rationale for analysing the data by calendar quarter is due to the gradual nature of prescribing changes and the fact that Irish GMS patients in receipt of medication (such as statins or ACEs) can receive three-months' worth of repeat prescriptions every time they visit their GP. The types of medications analysed are generally repeat rather than acute prescriptions. Consequently we consider

the use of data studied by calendar quarter to be consistent with the literature and appropriate in the Irish context. As such it does not rule out consideration of seasonal autocorrelation. Our methods section has been expanded to discuss this as follows and to take account of the matter of seasonal autocorrelation:

(p6): The time-scale used for the analyses of time series depends on the research question of interest (9). Calendar quarters (January-March, April-June, July-September, October-December) were used to aggregate the data consistent with other analyses of prescribing data using interrupted time series (10-12). The use of calendar quarters was deemed clinically appropriate: changes in prescribing patterns tend to be gradual and guidelines are not necessarily disseminated or actioned on the first day of each calendar month. Furthermore Irish GMS eligible patients in receipt of prescription medication can receive three-months' worth of repeat prescriptions per consultation with their GP.

(p8)

The potential for seasonal autocorrelation was also considered: in this context seasonal autocorrelation would mean that a given medication within a drug class is on average more or less likely to be prescribed than other drugs in the same class by virtue of the time of year. The PDI guidelines do not refer to any such clinical considerations (6) and we additionally hypothesised that seasonal autocorrelation would not be of statistical significance. This hypothesis was tested for each drug class by comparing the regression models which included Fourier terms to account for seasonality (9) and models without the seasonality terms. For each drug class seasonal autocorrelation was not of statistical significance and the seasonality terms were removed on the grounds of parsimony.

4- The authors did not use any reference group as a control to ensure that the observed changes were potentially contributed to the PDI. I understand that the PDI was a national initiative and hence was not possible to identify an intervention free-area, however, other approaches could have been used such as using a related but different outcome within the intervention group that would not be influenced by the intervention.

Response. As the reviewer rightly observes no obvious control group was available given that PDI guidelines were national. However we extended the analyses on the reviewer's recommendation as follows (p8):

The PDI guidelines were national guidelines and consequently no control groups were available with which to compare prescribing under the PDI. However, we constructed two reference groups using the drug classes beta-blockers and calcium channel blockers. These were drug classes for which PDI guidelines were launched in September 2016 (the preferred drugs being bisoprolol and amlodipine respectively) but for which no recommendations had been made when the PDI guidelines were launched for the other drug classes. Given that the earlier guidelines were launched within six months of each other two additional models were fitted: one examining prescribing of bisoprolol as the preferred beta-blocker over the study period, allowing for potential changes in prescribing when guidelines for PPIs/statins (April 2013) and SNRIs/SSRIs (April 2014) were disseminated, and one model examining prescribing of amlodipine as the preferred calcium channel blocker, allowing for potential changes in prescribing when guidelines for ACE inhibitors/ARBs (Sept 2013) and urology medications (October 2014) were issued.

(p12)

Beta-blockers and calcium channel blockers accounted for 3.58% (n=12,056,378) and 2.30% (n=7,753,755) of single-agent medications for GMS patients between 2011 and 2016, with the most commonly prescribed medications being bisoprolol (56.83% of all single-agent beta-blockers

(n=6,852,022)) and amlodipine (64.70% of all single-agent calcium channel blockers (n=5,016,348)), both of which were selected as preferred drugs in September 2016. There was a steady increase in prescribing of bisoprolol as the beta-blocker of choice and a consistent fall in prescribing of amlodipine within the calcium channel blockers over the study period. Effects associated with dissemination of the PDI guidelines for the other drug groups were non-significant at the 5% level.

Results

Through the whole result section, the authors described the changes as “modest” or “large” which is inappropriate. For instance, line 47, “a large reduction” to describe 0.98% reduction. All the changes in the first 3 months before and after the PDS described in Table 1 do not even reach 1%.

Response: We have revised the results section to refrain from use of the words modest/large, and either used the word small or simply reported “increase” or “decrease” or “some” change. What may be regarded as small to one reader might be considered modest to another regardless of the statistical significance.

Discussion

R2- Again, inappropriate use of “modest” to describe the PDS impact

Response: We have revised the discussion section to refrain from use of the words modest/large, and either used the word small or simply reported “increase” or “decrease”. It is very apparent that any changes in prescribing observed are very small at best.

R2- Line 21-24- here the authors show that factors other than the PDI were the reason behind the ACEIs and ER trends, and this could have been addressed, as I have explained earlier, by accounting for these other potential changes and by having a reference control group. Also, this can be further supported by the observed cost saving in ACEIs despite no change in prescribing trend after the PDI which explain that factors other than the PDI was the driving factor.

Response: licensing of mirabegron Jan 2013 and generic Duloxetine Sept 2014 have now been included in the models as discussed above.

R2- It is well known from the literature that passive and voluntary initiatives are mostly ineffective, and that multiple interventions are usually required to influence prescribing behavior (line 29-60), and this was the authors' argument. But in fact the authors default contradicts their study findings by showing that PDI, which is a passive and voluntary initiative, has a significant impact of changing prescribing patterns and induce cost saving.

Response:

We do not feel that the PDI has had a significant impact in changing prescribing patterns and inducing cost savings however it has to be acknowledged that some changes in prescribing have taken place. We have not described these changes as being significant in terms of prescription volume. The beginning of the section on policy implications states (p15):

The PDI has been developed to encourage evidence-based, cost-effective prescribing, but in view of the limited changes to date it has delivered only a small amount of potential cost savings in terms of the money spent on the prescription items.

R2- The authors mentioned the potential policy implications of their findings for Ireland, but cannot see any wider implications for the wider European and international community which would suggest the irrelevance of publishing this study in journals other than local Irish journals.

We have extended the policy implications discussion as follows:

Findings from this evaluation of the PDI in Ireland may be of interest to other countries which have implemented (e.g. Australia) or are considering preferred drug schemes or any intervention aimed at changing prescribing or clinical practice. The heterogeneity within our results illustrates that interventions developed using the same methodological framework may not necessarily yield comparable results even when launched concurrently.

The abstract has been extended (p2):

These findings are relevant where health services are seeking to develop more active prescribing interventions aimed at changing prescribing practice.

VERSION 2 – REVIEW

REVIEWER	Samantha Hollingworth School of Pharmacy The University of Queensland Brisbane, Australia
REVIEW RETURNED	03-Nov-2017

GENERAL COMMENTS	I am satisfied with the authors' responses to the reviewers' concerns. An earlier comment was about the figures – consider modifying figures so that they are legible in black and white. The authors say that this is the output that Stata provides (e.g. line figures). I suggest they extract the values and replot the graphs in Excel or some other software.
---

REVIEWER	Amanj Kurdi Strathclyde Institute of Pharmacy and Biomedical Science, Scotland, UK
REVIEW RETURNED	12-Nov-2017

GENERAL COMMENTS	Overall, the authors have addressed my comments appropriately; however, the following minor comments need to be considered Results line 24-15: I cannot see the effects of PDI on the other group in Table 3; also from Table 2 the effects of PDI were significant for most drug groups; clarify please Discussion line 28-30: how and why the findings from this study may be of interest to other countries? conclusion I think it is inappropriate to conclude that the observed changes in prescribing patterns for the six drug groups and the subsequent cost savings are due to the PDI guidelines because there were
--

	concurrent, significant changes in the prescribing of the reference groups as well as cost saving in ACEI cost despite no impact of PDI on its prescribing; these results, therefore, suggest that the observed changes were likely to be caused by factors other than the PDI.
--	---

VERSION 2 – AUTHOR RESPONSE

These responses are also available as a supplementary document.

Responses to Reviewers

Thanks to both of you for going through the revised PDI paper and for your comments.

Reviewer: 1

An earlier comment was about the figures – consider modifying figures so that they are legible in black and white. The authors say that this is the output that Stata provides (e.g. line figures). I suggest they extract the values and replot the graphs in Excel or some other software.

Response: We have revised the colour scheme/legends for both Figures and are satisfied that they are now clearly legible in both colour and in black and white.

Reviewer: 2

Results line 24-15: I cannot see the effects of PDI on the other group in Table 3; also from Table 2 the effects of PDI were significant for most drug groups; clarify please

Response: Table 3 only contains results from the analyses for both the reference groups-row1 beta blockers, row 2 calcium channel blockers. Two changes of level were included for both groups, with the second one year apart from the first. We have added extra footnotes to the table relating each change of level in the reference groups to dissemination of the appropriate PDI guidelines for the seven medicine groups analysed. As is noted none are significant at the 5% level.

Table 2: this only contains analyses relating to the 7 drug groups for which PDI guidelines were introduced between 2011 and 2016. Yes, changes in level were statistically significant in 6 of the 7 groups and that for extended release tolterodine/urology medications was non-significant. We have revised the text in the results (p11) to clarify:

In all medicine groups except urology, there was significant evidence of increases in prescribing of the preferred drugs immediately following dissemination of the PDI guidelines .

The wording regarding ACE inhibitors has been revised as follows:

There was a small increase in prescribing of the preferred ACE inhibitor (ramipril) immediately following the PDI guidelines (0.16%, 9%CI: 0.01 to 0.31, p=0.04), although subsequent increases per calendar quarter did not differ significantly at the 5% level from increases observed per calendar quarter prior to the PDI guidelines (p=0.08).

Discussion

line 28-30: how and why the findings from this study may be of interest to other countries?

Response: Given the increasing demand for services and increasing costs of health care provision, these findings are of interest to any country trying to find new ways of providing treatment that is both evidence-based and cost-effective, whether it is in relation to prescription costs or any form of treatment. They show that voluntary schemes aimed at changing clinical practice can be limited in their impact, and that when the same type of intervention or guidelines are issued in similar contexts similar results cannot be guaranteed. We have slightly revised the last paragraph in the discussion accordingly as follows:

Given the increasing demand for and costs associated with health-care provision world-wide, findings from this evaluation may be of interest to other countries seeking to provide treatment that is both evidence-based and cost-effective. This includes countries already implementing preferred drug schemes (e.g. Australia), those which are considering such schemes or indeed any intervention aimed at changing clinical practice. The results show that initiatives which are primarily voluntary in nature may be impactful but their impact can be limited and short-term. They also show that interventions launched concurrently and developed using the same methodological framework may not necessarily yield similar results.

conclusion

I think it is inappropriate to conclude that the observed changes in prescribing patterns for the six drug groups and the subsequent cost savings are due to the PDI guidelines because there were concurrent, significant changes in the prescribing of the reference groups as well as cost saving in ACEI cost despite no impact of PDI on its prescribing; these results, therefore, suggest that the observed changes were likely to be caused by factors other than the PDI.

Response: Given that there were no significant changes in prescribing in the reference groups associated with introduction of PDI guidelines in the other seven medicine groups (Table 3), that there were significant increases in prescribing of the preferred drugs following the PDI guidelines in six of the seven medicine groups (Table 2), that the combined costs savings in the two reference groups (which are commonly prescribed drugs) was €17k and that for the seven medicine groups €2.7 million, we feel it is inappropriate to conclude that, as a whole, the observed changes in prescribing and cost savings in the seven medicine groups considered were likely to be caused by other factors other than the PDI. We therefore are satisfied that the conclusion as stands is appropriate, with a slight rewording of the second sentence as follows:

Conclusions

Since the introduction of the PDI in 2013, there have been some cost savings across the PDI drug classes. However, more intensive implementation is needed if the PDI is to deliver the estimated €15million per year cost saving that was anticipated. Multifaceted interventions will be required to enhance the coverage and impact of the PDI so that these benefits can be realised.

Note: The paper has reached the maximum word limited allowed by the journal (4,000 words)

VERSION 3 – REVIEW

REVIEWER	Amanj Kurdi Strathclyde Institute of Pharmacy and Biomedical Science University of Strathclyde, Glasgow, Scotland, UK
REVIEW RETURNED	05-Dec-2017

GENERAL COMMENTS	I am happy with the response from the authors and they addressed the points adequately; but only minor comment is that still we cannot say the observed changes were totally due to the PDI guidelines; in the limitation it is needed to acknowledge that factors other than PDI guidelines might have contributed to the observed cost savings and changes in prescribing
---

REVIEWER	Samantha Hollingworth The University of Queensland, Australia
REVIEW RETURNED	16-Jan-2018

GENERAL COMMENTS	The authors provide a file called Responses_to_Reviewers_draft3. I could not see responses to my original comments (reproduced below). Please make it as easy as possible for the reviewers to see how you have addressed their comments (e.g. a table format with two columns is helpful). Original comments This is an interesting and important study examining the (rather disappointing) effects of the PDI to improve the quality use of medicines and reduce medicine costs. The manuscript was well written and logical. A line spacing of at least 1.5 would have improved readability. Abstract Line 7 Consider one sentence in objective how the PDI reduces costs. (Intro: The PDI recommends a single 'preferred drug' within a therapeutic drug class as the prescriber's drug of first choice.) Abstract says GMS is 40% of the population but later (S&L) you say 38% - be consistent Introduction. Line 6 Give an estimate of the GMS population □ you can estimate a cost per person. Clarify if the preferred drugs are cheaper than others in the class. Otherwise how does the HSE estimate the savings? Does the PDI it incorporate branded vs generic prescribing? P32 no need to capitalise medicine groups (even though they make an acronym) Methods Line 11 Citation for data source Line 22 generic medicine names are not capitalised (ramipril) I am not able to comment on the statistical analyses. Results P7 line 18 (and other places) use the term 'proportion' (percentage is the measure) P8 line 3 no need for brackets around the 95% CIs P8 line 5 absolute increases - in prescriptions? Figures – consider modifying figures so that they are legible in black and white (i.e. white, grey, stippled, black bars, etc.). The blue, red,
--

	and green bars can be hard to differentiate, even in colour. Figure 2 – at least use different symbol type to differentiate. The line figures (e.g. 2A) are also hard to differentiate. Discussion P 9 line 49 Australia also has a therapeutic group premium https://www.pbs.gov.au/browse/group-premium Please clarify if patents would have paid more, less, or the same if changed to the preferred medicine. P10 line 42 What was the most widely used statin - atorvastatin? What was the price difference of simvastatin compared to other statins in the group? Would there have been more HSE savings if the preferred medicine had been prescribed more widely (6.6% is very low for the preferred one especially seeing as statins are the most widely prescribed class of medicines!).
--	--

VERSION 3 – AUTHOR RESPONSE

All responses below are uploaded in a Word Document: Responses to reviewers_draft4, which may be easier to read.

Thanks to both reviewers for going through the revised PDI paper and for your comments.

Reviewer: 2

I am happy with the response from the authors and they addressed the points adequately; but only minor comment is that still we cannot say the observed changes were totally due to the PDI guidelines; in the limitation it is needed to acknowledge that factors other than PDI guidelines might have contributed to the observed cost savings and changes in prescribing

Response: We have added to the limitations sections the following sentence.

Although the changes in prescribing observed within the PDI medicine groups were not observed in the reference groups, there may be factors other than the PDI guidelines which have contributed to prescribing changes and the associated cost savings within the PDI drug groups.

Reviewer: 1

The authors provide a file called Responses_to_Reviewers_draft3. I could not see responses to my original comments (reproduced below).

Response: The responses to the original comments were submitted in the second iteration of the paper. As such only responses to further comments raised in the second version of the paper were appended to the submitted third version, on the understanding that the earlier responses had been considered and were sufficient. If there was any misunderstanding of the review process on our part, we apologise.

For the purposes of clarify, we are appending below all responses to all comments from both reviewers to all stages of the review process. We hope this will help show that we have found these helpful and have genuinely sought to incorporate all feedback into the paper. The tracked version appended to this submission includes all changes to the original paper regardless of at what stage the changes were made.

Responses to reviewers following original submission, accompanying draft 2.

Thank you to the editor and both reviewers for your helpful comments in the revision of this manuscript, and the opportunity to address these comments.

Editor Comments to Author:

- Please include the study design in the title. This is the preferred format of the journal.

Response: The title has been revised to: An evaluation of prescribing trends and patterns of claims within the Preferred Drugs Initiative in Ireland (2011-2016): an interrupted time-series study.

Reviewer: 1

R1: A line spacing of at least 1.5 would have improved readability.

Response: this has been changed to improve readability.

R1: Abstract Line 7 Consider one sentence in objective how the PDI reduces costs. (Intro: The PDI recommends a single 'preferred drug' within a therapeutic drug class as the prescriber's drug of first choice.)

Response: objective has been revised to read (p2): To examine the impact of the Preferred Drugs Initiative (PDI), an Irish health policy aimed at enhancing evidence-based cost-effective prescribing, on prescribing trends and the cost of prescription medicines across seven medication classes.

R1: Abstract says GMS is 40% of the population but later (S&L) you say 38% - be consistent.

Response: All figures have been revised to 40%.

R1: Introduction. Line 6 Give an estimate of the GMS population à you can estimate a cost per person.

Response: This has been now included on p10: The median number of items reimbursed per GMS patient was 63 (Interquartile Range (IQR) 13 to 246)) with a median total cost per patient of €905.75 (IQR €170.25 to €9,726.93).

R1: Clarify if the preferred drugs are cheaper than others in the class. Otherwise how does the HSE estimate the savings?

Response: Thank you for this comment: cost is one component to the recommendation of a preferred drug. Consequently the preferred drug may or may not necessarily be the cheapest drug within any

drug class. We do not have access to the calculations used by the HSE whereby they obtained their projected savings. The text has been revised to say (p4):

Although the preferred drugs may not necessarily be the least expensive medications within each drug class, it has been estimated that increased provision of the preferred drugs could save the HSE €15 million per year.(5)

R1: Does the PDI it incorporate branded vs generic prescribing?

Response: The issue of branded v generic prescribing is separate to the PDI. To-date reference prices have been set for some but not all generic drugs, including medicine groups not covered by the PDI. If a GMS patient wishes to receive a branded drug where a generic drug is available, they will be required to pay the price between the reference price and actual cost unless the prescriber has written "Do not substitute" on the prescription. The text has been amended to read (p4):

The regulations covering generic substitution of branded medications are separate to the PDI guidelines, with generic substitution of drugs implemented where possible unless there are clinical reasons for prescribing the branded medication.

R1: P32 no need to capitalise medicine groups (even though they make an acronym)

Response:we have corrected this in the text and graphs as appropriate

R1: Methods

Line 11 Citation for data source

Response: A citation for this has now been included.

R1: Line 22 generic medicine names are not capitalised (ramipril)

Response: This has been amended accordingly.

I am not able to comment on the statistical analyses.

R1: Results

P7 line 18 (and other places) use the term 'proportion' (percentage is the measure).

Response: Where percentages have been reported in the text or plotted on a graph, the text has been amended to percentage. The revisions of the models so that the outcome is percentage means that percentage is the appropriate scale on which to report results.

R1: P8 line 3 no need for brackets around the 95%CI's

Response: The formatting of the confidence intervals has been amended as follows:
lansoprazole (1.21%, 95%CI: 0.84% to 1.57%, p<0.001);

R1: P8 line 5 absolute increases - in prescriptions?

Response: As the outcome is now percentage coverage of PDI drug per class per quarter, the text refers to changes in the percentage coverage of the preferred drug within the class ie an increase in coverage from 20% to 21% is an increase of 1 %.

R1: Figures – consider modifying figures so that they are legible in black and white (i.e. white, grey, stippled, black bars, etc.). The blue, red, and green bars can be hard to differentiate, even in colour. Figure 2 – at least use different symbol type to differentiate. The line figures (e.g. 2A) are also hard to differentiate.

Response: We have revised the figures accordingly -Stata doesn't allow for striped/patterned bars in its settings.

R1: Discussion P 9 line 49 Australia also has a therapeutic group premium
<https://www.pbs.gov.au/browse/group-premium>

Response: We have inserted a sentence to this effect (p14);
Australia operates a therapeutic brand premium scheme, whereby a co-payment is required from patients when a prescriber has issued a drug within a drug class that is priced above the benchmark for drugs in that group.(23).

R1: Please clarify if patients would have paid more, less, or the same if changed to the preferred medicine.

Response: GMS patients in Ireland have their prescription charges paid directly by the State, with a patient-levy of €2.50 for each item dispensed, up to a maximum of €25 per month. This is the same regardless of whether it is a preferred or non-preferred drug. We have inserted in the text (p4)

Currently GMS patients in Ireland have their prescription charges paid directly by the State, with a patient-levy of €2.50 for each item dispensed, up to a maximum of €25 per month.

The issuing of preferred drugs is voluntary and no incentives are given to prescribers to issue the preferred drug instead of others from within the same therapeutic drug class, with the patient levy remaining unaltered irrespective of preferred- or non-preferred drug status.

R1: P10 line 42 What was the most widely used statin - atorvastatin? What was the price difference of simvastatin compared to other statins in the group? Would there have been more HSE savings if the preferred medicine had been prescribed more widely (6.6% is very low for the preferred one especially seeing as statins are the most widely prescribed class of medicines!).

Response: The most commonly prescribed statin currently in Ireland is atorvastatin (57.3% of all single-agent Statins between 2011-2016). simvastatin is the cheapest Statin, rosuvastatin the most expensive. The amount of savings will depend both on any increased volume of the preferred drug and the price differential between the preferred drug and what it has been substituted for. We have clarified this in the text as follows (p15):

If efforts are to be enhanced, the energies need to focus on medicine groups which are large volume (e.g. PPIs and statins) and/or where there is considerable variation between the least and most expensive licensed medications (e.g. SNRIs).

Reviewer: 2

Reviewer Name: Amanj Kurdi

Institution and Country: Strathclyde Institute of Pharmacy and Biomedical Science, University of Strathclyde, Scotland, UK Please state any competing interests or state 'None declared': None to declare

Please leave your comments for the authors below See the attached file please
Comments

Manuscript title: An evaluation of prescribing trends and patterns of claims within the Preferred Drugs Initiative in Ireland: 2011-2016

The study examined the impact of the Preferred Drugs Initiative (PDI), an Irish health policy aimed at reducing the cost of prescription medicines using segmented regression. Although the study looks interesting, it has major methodological limitations which make it unacceptable for publication at such.

R2: Background

- 1- Line 8- Not clear who are eligible for the PCR
- 2- Line 9- what does "means testing" mean?

Response: The text has been revised for clarity as follows (p4):

This (ie the PCRS GMS scheme) is the largest community drug scheme in Ireland, providing access to free or minimal cost health care for patients whose household income falls below the eligibility threshold specified by the Irish Government, as well as the majority of people aged ≥ 70 years (approximately 95%) where a higher income threshold applies.

R2: 3- Line 31- the authors mentioned that an evaluation report has been published in September 2016 for the first ten drug classes, but they did not mention the content of the report. It looks like the report has evaluated this PDI; therefore, it is not clear what this study will add, i.e, I cannot see clear and strong justification and rationale for the study.

Response: individual reports detailing the rationale for the choice of preferred drug within each class have been issued on an ongoing basis between April 2013 and Sept 2016, but to date no single report has been published which seeks to evaluate changes in prescribing which may have occurred with implementation of the PDI. Consequently this is the rationale for this research. The text has been revised to clarify this as follows (p5)

As of September 2016 reports detailing the rationale behind the choice of preferred drugs have been published for the first ten therapeutic drug classes covered by the Initiative.

An additional sentence has been added.

There has been no evaluation of changes in prescribing following the introduction of the PDI to date.

R2 Methods

The study method suffers major limitations:

Data

Line 11- it is not clear how selecting 2011 to 2016 as the study period will avoid confounding because other pharmaceutical policy changes could still happen within 2011-2016

Response. Thank you for this comment, this was a poorly worded and incorrect sentence. 2011-2016 was chosen as the study period so that across the 7 medicine groups as a whole the average time studied pre-PDI guidelines would be similar to the average studied post-PDI guidelines. We have revised this sentence as follows (p6):

This study period provided an average of three years of claims data both before and after the guidelines across the medication groups considered.

PDI

Line 27- why the study population was limited to >18 years old? This is not suitable as the PDI targeted prescriptions which would be prescribed to any age group.

Responses: The majority of the medication classes considered (e.g. statins, ACE, ARBs) are not commonly prescribed to children. The PDI was developed specifically to enhance cost-effective evidence-based prescribing among adults. The text has been clarified to say (p6):

; the PDI is primarily aimed at the treatment of adults in the general population.

Analytical method

R2- The authors did not tested and accounted for auto-correlations as the study outcomes measured repeatedly each quarter are very likely to be auto correlated which will ultimately bias the study results

Response. Thank you for this comment which was a very important consideration. Autocorrelation coefficients were examined which showed there was significant autocorrelation at one lag only, and so the models were re-estimated to include this. In doing so we switched from logistic to linear regression, given that the denominators (number of drugs per class per quarter) are large. The text states this as follows (p7):

Examination of the autocorrelation and partial autocorrelation coefficients showed that there was significant residual autocorrelation between adjacent calendar quarters (but not between non-adjacent quarters) in each drug group, and this was incorporated into the models using Prais–Winsten regression (16).

R2- The analysis did not account for the impact of any other policies which might affect the study outcomes such as changes in drug prices, new treatment guidelines or when a drug in the class became off patent. For instance, the availability of generic candesartan, or change in hyperlipidemia management guidelines after the availability of generic atorvastatin which could have a potential impact on which drug to choose within the class.

Response:

We have considered this and extended two medicine groups as detailed below in the text (p7):

More than one change of level can be incorporated into any interrupted time series where this is relevant to the research question (13, 15). It was not feasible to include changes in the price of drugs in these models given the large number of drugs considered. Across the drug classes all drugs were licensed and available in Ireland between 2011 and 2016, and all generics were licensed prior to the study period, the key exceptions being the licensing of generic duloxetine in April 2015 and the licensing of mirabegron in January 2013. These two events were incorporated into the analyses for SNRIs and urology medications respectively.

R2- Line 40- to perform segmented analysis properly, at least 12 time points are needed before and after the intervention to allow adjustment for seasonal autocorrelation; therefore, using calendar quarters to reduce the number of time-points is not appropriate.

Response:

We have considered this matter very carefully. All time series have the potential to exhibit seasonal correlation, however this does not mean that all time-series have to be analysed using month as the unit of time just to address seasonal autocorrelation and other time scales are possible. Wagner et al.(2002) states that that time series are required to summarise outcomes “at regular, evenly spaced intervals”. Consequently when they state (p305) “Detecting seasonality requires baseline series that span enough periods to detect these cyclic patterns...To estimate seasonal autocorrelation, the auto-regression model needs to evaluate correlations between error terms separated by 12 months” we interpret their sentence “Accounting for seasonally correlated errors usually requires at least 24 monthly data points” not to mean that data has to be analysed on a monthly time scale, but rather, where data is analysed on a monthly time scale (as is the case in the data they are presenting and analysing), there will be a full calendar year of data before the interruption and a full calendar year of data afterwards. We appreciate this sentence could be read either way.

Other papers make the matter more obvious: Bernal et al. (2017) state that methods for controlling for seasonality can include: “ a model stratified by the calendar month (or any other period)”. Bhaskan et al. (2013) show how to construct Fourier series to include seasonality in a time-series, “by creating a degrees variable for time divided by the number of time points in a year”. Therefore we understand the literature to say that the important feature is there is a full-calendar year pre/post any interruption, rather than that data has to be analysed on a monthly basis. Outcomes may not necessarily be expected to vary by month and they could vary by season.

The literature is not consistent regarding the number of time points to be included in interrupted time-series pre/post changes of level. Wagner et al.(2002) recommends 12 points on either side of the interruption, but go on to say that this is not based on estimates of statistical power but relates to having 12 monthly measures (i.e. a full calendar year) pre/post the interruption due to concerns over seasonality. Bernal et al. (2010) states that “power increases with the number of time points, but it is not always preferable to have more data points where historical trends have changed substantively”. They do not state a minimum number of points pre/post interruptions and suggest inspection of the data to get a visual impression on any trends. Flodgren & Oddgard-Jensen (2013) state you need “ at least three data points before and three data points after the intervention”.

In Penfold et al.'s teaching paper on interrupted time series (2013) there are ten quarterly time-points before the intervention and nine afterwards. We have at least nine time points on either side of the PDI guidelines in each drug group where there was only one change of level. Where a second change of level was incorporated as per the reviewer's helpful recommendations, we have still met Flodgren & Oddgard-Jensen's recommendation above. All our time-series were subject to sensitivity analysis to explore whether changing the number of time-points before and after the interruptions resulted in any substantive changes to our results: they did not.

Our rationale for analysing the data by calendar quarter is due to the gradual nature of prescribing changes and the fact that Irish GMS patients in receipt of medication (such as statins or ACEs) can receive three-months' worth of repeat prescriptions every time they visit their GP. The types of medications analysed are generally repeat rather than acute prescriptions. Consequently we consider the use of data studied by calendar quarter to be consistent with the literature and appropriate in the Irish context. As such it does not rule out consideration of seasonal autocorrelation. Our methods section has been expanded to discuss this as follows and to take account of the matter of seasonal autocorrelation:

(p6): The time-scale used for the analyses of time series depends on the research question of interest (9). Calendar quarters (January-March, April-June, July-September, October-December) were used to aggregate the data consistent with other analyses of prescribing data using interrupted time series (10-12). The use of calendar quarters was deemed clinically appropriate: changes in prescribing patterns tend to be gradual and guidelines are not necessarily disseminated or actioned on the first day of each calendar month. Furthermore Irish GMS eligible patients in receipt of prescription medication can receive three-months' worth of repeat prescriptions per consultation with their GP.

(p8)

The potential for seasonal autocorrelation was also considered: in this context seasonal autocorrelation would mean that a given medication within a drug class is on average more or less likely to be prescribed than other drugs in the same class by virtue of the time of year. The PDI guidelines do not refer to any such clinical considerations (6) and we additionally hypothesised that seasonal autocorrelation would not be of statistical significance. This hypothesis was tested for each drug class by comparing the regression models which included Fourier terms to account for seasonality (9) and models without the seasonality terms. For each drug class seasonal autocorrelation was not of statistical significance and the seasonality terms were removed on the grounds of parsimony.

4- The authors did not use any reference group as a control to ensure that the observed changes were potentially contributed to the PDI. I understand that the PDI was a national initiative and hence was not possible to identify an intervention free-area, however, other approaches could have been used such as using a related but different outcome within the intervention group that would not be influenced by the intervention.

Response. As the reviewer rightly observes no obvious control group was available given that PDI guidelines were national. However we extended the analyses on the reviewer's recommendation as follows (p8):

The PDI guidelines were national guidelines and consequently no control groups were available with which to compare prescribing under the PDI. However, we constructed two reference groups using the drug classes beta-blockers and calcium channel blockers. These were drug classes for which PDI guidelines were launched in September 2016 (the preferred drugs being bisoprolol and amlodipine respectively) but for which no recommendations had been made when the PDI guidelines were launched for the other drug classes. Given that the earlier guidelines were launched within six months of each other two additional models were fitted: one examining prescribing of bisoprolol as the preferred beta-blocker over the study period, allowing for potential changes in prescribing when guidelines for PPIs/statins (April 2013) and SNRIs/SSRIs (April 2014) were disseminated, and one model examining prescribing of amlodipine as the preferred calcium channel blocker, allowing for potential changes in prescribing when guidelines for ACE inhibitors/ARBs (Sept 2013) and urology medications (October 2014) were issued.

(p12)

Beta-blockers and calcium channel blockers accounted for 3.58% (n=12,056,378) and 2.30% (n=7,753,755) of single-agent medications for GMS patients between 2011 and 2016, with the most commonly prescribed medications being bisoprolol (56.83% of all single-agent beta-blockers (n=6,852,022)) and amlodipine (64.70% of all single-agent calcium channel blockers (n=5,016,348)), both of which were selected as preferred drugs in September 2016. There was a steady increase in prescribing of bisoprolol as the beta-blocker of choice and a consistent fall in prescribing of amlodipine within the calcium channel blockers over the study period. Effects associated with dissemination of the PDI guidelines for the other drug groups were non-significant at the 5% level.

Results

Through the whole result section, the authors described the changes as “modest” or “large” which is inappropriate. For instance, line 47, “a large reduction” to describe 0.98% reduction. All the changes in the first 3 months before and after the PDS described in Table 1 do not even reach 1%.

Response: We have revised the results section to refrain from use of the words modest/large, and either used the word small or simply reported “increase” or “decrease” or “some” change. What may be regarded as small to one reader might be considered modest to another regardless of the statistical significance.

Discussion

R2- Again, inappropriate use of “modest” to describe the PDS impact

Response: We have revised the discussion section to refrain from use of the words modest/large, and either used the word small or simply reported “increase” or “decrease”. It is very apparent that any changes in prescribing observed are very small at best.

R2- Line 21-24- here the authors show that factors other than the PDI were the reason behind the ACEIs and ER trends, and this could have been addressed, as I have explained earlier, by accounting for these other potential changes and by having a reference control group. Also, this can be further supported by the observed cost saving in ACEIs despite no change in prescribing trend after the PDI which explain that factors other than the PDI was the driving factor.

Response: licensing of mirabegron Jan 2013 and generic Duloxetine Sept 2014 have now been included in the models as discussed above.

R2- It is well known from the literature that passive and voluntary initiatives are mostly ineffective, and that multiple interventions are usually required to influence prescribing behavior (line 29-60), and this was the authors' argument. But in fact the authors default contradicts their study findings by showing that PDI, which is a passive and voluntary initiative, has a significant impact of changing prescribing patterns and induce cost saving.

Response:

We do not feel that the PDI has had a significant impact in changing prescribing patterns and inducing cost savings however it has to be acknowledged that some changes in prescribing have taken place. We have not described these changes as being significant in terms of prescription volume. The beginning of the section on policy implications states (p15):

The PDI has been developed to encourage evidence-based, cost-effective prescribing, but in view of the limited changes to date it has delivered only a small amount of potential cost savings in terms of the money spent on the prescription items.

R2- The authors mentioned the potential policy implications of their findings for Ireland, but cannot see any wider implications for the wider European and international community which would suggest the irrelevance of publishing this study in journals other than local Irish journals.

We have extended the policy implications discussion as follows:

Findings from this evaluation of the PDI in Ireland may be of interest to other countries which have implemented (e.g. Australia) or are considering preferred drug schemes or any intervention aimed at changing prescribing or clinical practice. The heterogeneity within our results illustrates that interventions developed using the same methodological framework may not necessarily yield comparable results even when launched concurrently.

The abstract has been extended (p2):

These findings are relevant where health services are seeking to develop more active prescribing interventions aimed at changing prescribing practice.

Responses to reviewers following draft 2, accompanying draft 3.

Thanks to both of you for going through the revised PDI paper and for your comments.

Reviewer: 1

An earlier comment was about the figures – consider modifying figures so that they are legible in black and white. The authors say that this is the output that Stata provides (e.g. line figures). I suggest they extract the values and replot the graphs in Excel or some other software.

Response: We have revised the colour scheme/legends for both Figures and are satisfied that they are now clearly legible in both colour and in black and white.

Reviewer: 2

Results line 24-15: I cannot see the effects of PDI on the other group in Table 3; also from Table 2 the effects of PDI were significant for most drug groups; clarify please

Response: Table 3 only contains results from the analyses for both the reference groups-row1 beta blockers, row 2 calcium channel blockers. Two changes of level were included for both groups, with the second one year apart from the first. We have added extra footnotes to the table relating each change of level in the reference groups to dissemination of the appropriate PDI guidelines for the seven medicine groups analysed. As is noted none are significant at the 5% level.

Table 2: this only contains analyses relating to the 7 drug groups for which PDI guidelines were introduced between 2011 and 2016. Yes, changes in level were statistically significant in 6 of the 7 groups and that for extended release tolterodine/urology medications was non-significant. We have revised the text in the results (p11) to clarify:

In all medicine groups except urology, there was significant evidence of increases in prescribing of the preferred drugs immediately following dissemination of the PDI guidelines .

The wording regarding ACE inhibitors has been revised as follows:

There was a small increase in prescribing of the preferred ACE inhibitor (ramipril) immediately following the PDI guidelines (0.16%, 9%CI: 0.01 to 0.31, p=0.04), although subsequent increases per calendar quarter did not differ significantly at the 5% level from increases observed per calendar quarter prior to the PDI guidelines (p=0.08).

Discussion

line 28-30: how and why the findings from this study may be of interest to other countries?

Response: Given the increasing demand for services and increasing costs of health care provision, these findings are of interest to any country trying to find new ways of providing treatment that is both evidence-based and cost-effective, whether it is in relation to prescription costs or any form of treatment. They show that voluntary schemes aimed at changing clinical practice can be limited in their impact, and that when the same type of intervention or guidelines are issued in similar contexts similar results cannot be guaranteed. We have slightly revised the last paragraph in the discussion accordingly as follows:

Given the increasing demand for and costs associated with health-care provision world-wide, findings from this evaluation may be of interest to other countries seeking to provide treatment that is both evidence-based and cost-effective. This includes countries already implementing preferred drug schemes (e.g. Australia), those which are considering such schemes or indeed any intervention aimed at changing clinical practice. The results show that initiatives which are primarily voluntary in nature may be impactful but their impact can be limited and short-term. They also show that interventions launched concurrently and developed using the same methodological framework may not necessarily yield similar results.

conclusion

I think it is inappropriate to conclude that the observed changes in prescribing patterns for the six drug groups and the subsequent cost savings are due to the PDI guidelines because there were concurrent, significant changes in the prescribing of the reference groups as well as cost saving in ACEI cost despite no impact of PDI on its prescribing; these results, therefore, suggest that the observed changes were likely to be caused by factors other than the PDI.

Response: Given that there were no significant changes in prescribing in the reference groups associated with introduction of PDI guidelines in the other seven medicine groups (Table 3), that there were significant increases in prescribing of the preferred drugs following the PDI guidelines in six of the seven medicine groups (Table 2), that the combined costs savings in the two reference groups (which are commonly prescribed drugs) was €17k and that for the seven medicine groups €2.7 million, we feel it is inappropriate to conclude that, as a whole, the observed changes in prescribing and cost savings in the seven medicine groups considered were likely to be caused by other factors other than the PDI. We therefore are satisfied that the conclusion as stands is appropriate, with a slight rewording of the second sentence as follows:

Conclusions

Since the introduction of the PDI in 2013, there have been some cost savings across the PDI drug classes. However, more intensive implementation is needed if the PDI is to deliver the estimated €15million per year cost saving that was anticipated. Multifaceted interventions will be required to enhance the coverage and impact of the PDI so that these benefits can be realised.

Note: The paper has reached the maximum word limited allowed by the journal (4,000 words)

Responses to reviewers following draft 3, accompanying draft 4.

These are reported at the beginning of this document.

VERSION 4 – REVIEW

REVIEWER	Samantha Hollingworth University of Queensland
REVIEW RETURNED	19-Jan-2018
GENERAL COMMENTS	All comments were addressed well. There was some confusion about the response versions but am now satisfied. Well done.